

# Per-object systematics using deep-learned calibration

Gregor Kasieczka[1], Michel Luchmann[2*], Florian Otterpohl[1] and Tilman Plehn[2]

**1** Institut für Experimentalphysik, Universität Hamburg, Germany
**2** Institut für Theoretische Physik, Universität Heidelberg, Germany

⋆ luchmann@thphys.uni-heidelberg.de

## Abstract

We show how to treat systematic uncertainties using Bayesian deep networks for regression. First, we analyze how these networks separately trace statistical and systematic uncertainties on the momenta of boosted top quarks forming fat jets. Next, we propose a novel calibration procedure by training on labels and their error bars. Again, the network cleanly separates the different uncertainties. As a technical side effect, we show how Bayesian networks can be extended to describe non-Gaussian features.

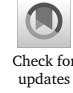

## Content

## 1 Introduction

Modern methods of machine learning are becoming a crucial tool in experimental and theoretical particle physics. An especially active field in this direction is subjet physics and jet

tagging [1], where multi-variate analyses of high-level observables are being replaced with deep neural networks working on low-level inputs. Early applications of deep learning techniques in LHC physics rely on image recognition of jet images [2, 3]. Their main challenge is to combine calorimeter and tracking information, motivating graph convolutional networks and point clouds [4]. Established benchmarks processes for these methods include quark-gluon discrimination [5–10], flavor tagging [11], $W$-tagging [12–15], Higgs-tagging [16, 17], or top-tagging [14, 15, 18–25]. By now we can consider top jet classification at the level of tagging performance as essentially solved [26, 27]. This gives us room to consider question beyond the performance, for instance what the networks are learning, how they can be visualized, how robust they are, how we can control the uncertainties, and how machine learning methods affect typical LHC analyses structurally.

One open question is driven by particle physics' obsession with error bars: how do we quantify the different uncertainties in analyses using neural networks [28–31]? This question is related to visualization [32], understanding the relevant physics features [33–37], and weakly supervised learning approaches [38–45] — all combined under the general theme of explainable AI. In LHC physics we have the advantage of excellent Monte Carlo simulations and full control of the experimental setup. This allows us to define and control different sources uncertainties very precisely. If we accept that a neural network is just a function relating training data to an output there exist (at least) two main kinds of uncertainties:

1. first, labelled training data comes with statistical and systematic uncertainties, where we define the former as uncertainties which vanish with more training data. The systematic uncertainties can be Gaussian or include shifts, depending on their sources. Unstable network training also belongs to this category of training-induced uncertainties [28];

2. second, on the test data or analysis side we also encounter statistical and systematic uncertainties. When we include an inference or any kind of analysis we also encounter model or theory uncertainties [29]. For these uncertainties it is crucial that we ensure our analysis outcome is conservative.

In a previous paper [28] we have shown how Bayesian classification networks can track uncertainties and provide jet-by-jet error bars for the tagging output. Such a Bayesian network can supplement a probabilistic classification output of '60% signal' with an error estimate of the kind '$(60 \pm 10)$% signal' for a given jet. This kind of jet-by-jet information exceeds what is available from standard LHC classification tools. In principle, this approach covers both, statistical errors from the size of the training sample and systematic uncertainties for instance from the calibration of the training sample. However, our quantitative analysis of Bayesian top taggers encountered practical limitations, for instance that the jet energy scale simultaneously affects the central value and the error bar of the probabilistic output. A similar study of uncertainties just appeared for a matrix element regression task [46].

In this follow-up study we look at this problem from a slightly different angle, now defining the *regression task* of extracting the energy of a tagged top quark inside a fat jet. Again, we translate statistical and systematic uncertainties from the training sample to the test output. The Bayesian network, introduces in Sec. 2, allows us to construct a per-jet probability distribution function over possible top momenta, or $p(p_t|\text{fat jet})$. The main advantage of using the regression task as example is that it does not enforce a closed interval for the network output and hence removes the correlation between central value and error estimate in the network output. We use this advantage to cleanly separate effects from the finite size of the training sample and from the stochastic nature of the training sample in Sec. 4.

In Sec. 5 the stochastic uncertainty leads us to a discussion of systematics in the sense of training-related uncertainties which do not shrink with more training data. Our regression task naturally leads us to developing a framework to calibrate deep network taggers and account

for uncertainties in the training sample. We find that a straightforward treatment should be based on smearing the momentum labels in the training sample. It directly accounts for the uncertainties in the underlying measurements of the calibration sample and treats them as an additional systematic effect on the top momentum measurement. As before, the Bayesian network allows us to cleanly separate all different sources of uncertainty.

Our simple application serves as an example how we can use Bayesian networks to define statistical and systematic uncertainties coming from the training sample and affecting the network output. These error bars are defined jet by jet, or event by event, giving us more control than standard methods do. Training on smeared labels allows us to implement energy calibration in a straightforward and automized manner. While our modelling of uncertainties on the reference measurements for calibration is simplified, our approach can be extended in a straightforward manner. For instance, the effect of different jet algorithms or different Monte Carlo simulations can be implemented as a non-Gaussian contribution to the label smearing. The key observation is that Bayesian networks allow us to quote uncertainties from all kinds of statistical and systematic limitations of the labelled training data.

## 2 Bayesian regression

While standard neural networks adapt a set of weights $\omega$ to describe a general function based on some kind of training, Bayesian networks learn weight distributions [47–52]. Sampling over those $\omega$-distributions gives us access to uncertainties in the network output, induced by limitations of the training data. After studying the effect of limited training statistics on jet classification [28], we now generalize our approach to include limited training statistics as well as the systematic effects from stochastic or smeared training data.

As an example, we want to extract the transverse momentum $p_T$ of a hadronically decaying top quark from a fat top jet. If we define $p(p_T|j)$ as the probability over possible $p_T$ values for a given top jet, $j$, we can extract the mean value as:

$$\langle p_T \rangle = \int dp_T \, p_T \, p(p_T|j) \,. \tag{1}$$

For a Bayesian network $p(p_T|j)$ is generated by sampling over the trained weight distributions $p(\omega|M)$,

$$p(p_T|j) = \int d\omega \, p(p_T|\omega, j) \, p(\omega|M) \,, \tag{2}$$

where $M$ is the training data set. Obviously, we do not know the closed form of $p(\omega|M)$. In the sense of a distribution [53], the network training will approximate it with the learned function $q(\omega)$,

$$p(p_T|j) = \int d\omega \, p(p_T|\omega, j) \, p(\omega|M) \approx \int d\omega \, p(p_T|\omega, j) \, q(\omega) \,. \tag{3}$$

If we exchange the two integrals, the mean transverse momentum becomes

$$\langle p_T \rangle \equiv \int d\omega \, q(\omega) \langle p_T \rangle_\omega \quad \text{with} \quad \langle p_T \rangle_\omega = \int dp_T \, p_T \, p(p_T|\omega, j) \,. \tag{4}$$

Correspondingly, the variance of the $p_T$ extraction can be extracted as

$$
\begin{aligned}
\sigma_{\text{tot}}^2 &= \langle (p_T - \langle p_T \rangle)^2 \rangle \\
&= \int d\omega \, q(\omega) \big[ \langle p_T^2 \rangle_\omega - 2 \langle p_T \rangle \langle p_T \rangle_\omega + \langle p_T \rangle^2 \big] \\
&= \int d\omega \, q(\omega) \big[ \langle p_T^2 \rangle_\omega - \langle p_T \rangle_\omega^2 + (\langle p_T \rangle_\omega - \langle p_T \rangle)^2 \big] \equiv \sigma_{\text{stoch}}^2 + \sigma_{\text{pred}}^2 \, .
\end{aligned}
\tag{5}
$$

This is the critical step which allows us to identify two contributions to the jet-wise uncertainty from the Bayesian network. First, a finite $\sigma_{\text{stoch}}$ occurs without even sampling the network weights, so it describes a systematic effect from the stochastic nature of the training sample,

$$
\begin{aligned}
\sigma_{\text{stoch}}^2 \equiv \langle \sigma_{\text{stoch},\omega}^2 \rangle &= \int d\omega \, q(\omega) \, \sigma_{\text{stoch},\omega}^2 \\
&= \int d\omega \, q(\omega) \big[ \langle p_T^2 \rangle_\omega - \langle p_T \rangle_\omega^2 \big] \, .
\end{aligned}
\tag{6}
$$

Second, $\sigma_{\text{pred}}$ is defined in terms of the $\omega$-integrated expectation value $\langle p_T \rangle$, so there does not exist an $\omega$-dependent version,

$$
\sigma_{\text{pred}}^2 = \int d\omega \, q(\omega) (\langle p_T \rangle_\omega - \langle p_T \rangle)^2 \, .
\tag{7}
$$

Only this second contribution will vanish in the limit of an infinitely large training sample, because in that case the network weight distributions become delta distributions. We will discuss the nature of these two contributions in detail in our analysis.

The two contributions to $\sigma_{\text{tot}}$ can also be identified in the loss function. The standard approach for Bayesian networks is to start with Eq.(3) implemented as a Kullback-Leibler divergence,

$$
\begin{aligned}
\text{KL}[q(\omega), p(\omega|M)] &= \int d\omega \, q(\omega) \, \log \frac{q(\omega)}{p(\omega|M)} \\
&= \int d\omega \, q(\omega) \, \log \frac{q(\omega) p(M)}{p(M|\omega) p(\omega)} \\
&= \underbrace{\text{KL}[q(\omega), p(\omega)] - \int d\omega \, q(\omega) \, \log p(M|\omega)}_{\equiv L_{\text{KL}}} + \log p(M) \int d\omega \, q(\omega) \, .
\end{aligned}
\tag{8}
$$

In this derivation we use Bayes' theorem. The prior $p(\omega)$ describes the model parameters before training. The model evidence $p(M)$ guarantees the correct normalization of $p(\omega|M)$. Turning Eq.(8) into a loss function we can omit it just as the normalization condition for $q(\omega)$. The relevant loss function of the Bayesian network, $L_{\text{KL}}$, then consists of two terms, the regularization for $q(\omega)$ in reference to the prior $p(\omega)$ and the likelihood $p(M|\omega)$, which we can work with in a frequentist sense. For a Gaussian prior the regularization term becomes the standard L2-regularization.

For illustration purposes or to improve the numerical performance we can now make a set of assumptions. In Ref. [28] we have shown, by varying priors over several order of magnitude, that assuming a Gaussian prior $p(\omega)$ had no visible effect on the network output. To get analytic control, we can approximate the likelihood $p(M|\omega)$ as Gaussian,

$$
\log p(M|\omega) \approx -\frac{\left( p_T^{\text{truth}} - \langle p_T \rangle_\omega \right)^2}{2\sigma_{\text{stoch},\omega}^2} - \frac{1}{2} \log \sigma_{\text{stoch},\omega}^2 + \text{const.} \, ,
\tag{9}
$$

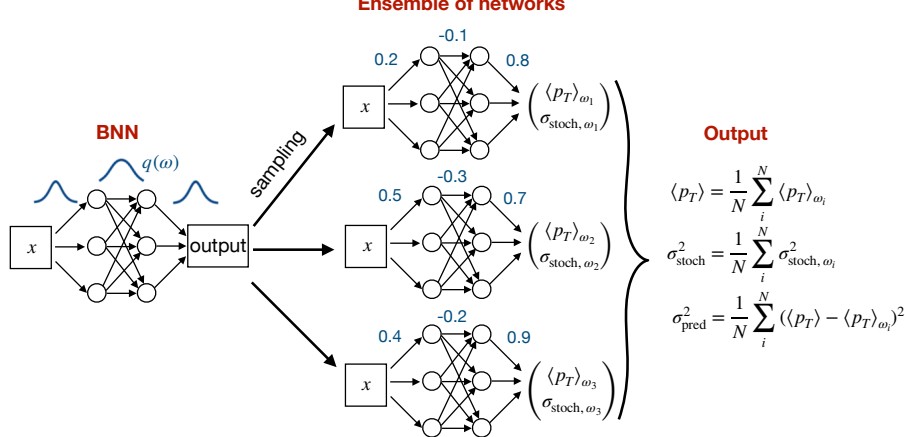

Figure 1: Illustration of our Bayesian network setup. The Bayesian network provides us with an uncertainty estimate for a single input jet $x$.

where $p_T^{\text{truth}}$ is the truth label provided by the training data set $M$. The width of this Gaussian corresponds to a systematic uncertainty, so we identify it with $\sigma_{\text{stoch},\omega}$. The loss function

$$L_{\text{KL}} \approx \text{KL}[q(\omega), p(\omega)] + \int d\omega \, q(\omega) \left[ \frac{\left(p_T^{\text{truth}} - \langle p_T \rangle_\omega\right)^2}{2\sigma_{\text{stoch},\omega}^2} + \frac{1}{2}\log\sigma_{\text{stoch},\omega}^2 \right], \qquad (10)$$

now has to be minimized with respect to the parameters of $q(\omega)$. Because we have assumed $q(\omega)$ to be Gaussian that gives us two trainable parameters per weight, and our neural network gives

$$\text{NN}(\omega) = \begin{pmatrix} \langle p_T \rangle_\omega \\ \sigma_{\text{stoch},\omega} \end{pmatrix} \qquad (11)$$

per jet. To extract the per-jet probability distribution $p(p_T|x)$ following Eq.(3), we usually rely on Monte Carlo integration by sampling weights from the weight distributions. As in Eq.(10) we assume that $p(p_T|\omega, x)$ is a Gaussian with the above-defined mean $\langle p_T \rangle_\omega$ and width $\sigma_{\text{stoch},\omega}$. Moreover, for large training statistics the distribution $q(\omega)$ should become narrow. According to Eq.(7) the effect of a finite width of $q(\omega)$ can be tracked by $\sigma_{\text{pred}}$, so in the limit $\sigma_{\text{pred}} \ll \sigma_{\text{stoch}}$ we can approximate $p(p_T|x)$ as a Gaussian with weight-independent mean $\langle p_T \rangle$ and width $\sigma_{\text{stoch}}$. This network structure is illustrated in Fig. 1.

## 3 Data set and network

The correct and precise reconstruction of the momentum of tagged top quarks is important for instance in top resonance searches and has influenced the design of many top taggers [54]. Our data set is therefore similar to standard top tagging references, with some modifications which simplify our regression task. We generate a sample of $R = 1.2$ top jets in the range $p_{T,t}^{\text{truth}} = 400 \dots 1000$ GeV with PYTHIA [55] at 14 TeV collider energy and the standard ATLAS card for DELPHES [56]. We always neglect multi-parton interactions and always include final state radiation. Given initial state radiation we work with two event samples, one with ISR switched on and one with ISR switched off. We require the jets to be central $|\eta_j| < 2$ and truth-matched in the sense that each fat jet has to have a top quark within the jet area. These

settings essentially correspond to the public top tagging data set from Refs. [20] and [27]. The difference to the standard tagging reference sets is that we flatten our data set in $p_{T,t}^{\text{truth}}$, such that even accounting for bin migration effects we can safely assume that in the fat jet momentum the sample is flat for $p_{T,j} = 500 \dots 800$ GeV.

The final result of our Bayesian network will be a probability distribution over possible $p_{T,t}$ values for a given jet. For our labelled data we know the corresponding $p_{T,t}^{\text{truth}}$. However, the fact that we will modify this truth label as part of the calibration training makes it the less attractive option to organize our samples. The closest alternative observable is the momentum of the fat jet, so we can think of $p_{T,j}$ as representing the complete fat jet input to the network. So unless explicitly mentioned we train our networks on a large data set defined in terms of the fat jet momentum,

$$p_{T,j} = 400 \dots 1000 \text{ GeV} \qquad \text{(training sample)} . \tag{12}$$

Whenever we need a homogeneous sample without boundary effects we choose a narrow test sample with

$$p_{T,j} = 600 \dots 620 \text{ GeV} \qquad \text{(narrow test sample)} . \tag{13}$$

The data format for the fat jet information is a $p_T$-ordered list of up to 200 constituent 4-vectors ($\vec{p}$ and $E$) with ISR and 100 constituents without. Our total sample size is 2.2M jets without ISR, of which we use 400k jet for validation and testing, each. The training size is varied throughout our analysis.

Our regression network is a simple 5-layer fully connected dense network. Its first two layers each consist of 100 units, the next two 50 units, followed by a 2-unit output layer, unless mentioned otherwise. For the prior we choose a Gaussian around zero and with width 0.1. We have confirmed that our results are width-independent over a wide range [28]. The typical sizes and widths of the weights depends on the input data. The input is a flattened set of 4-vectors where we re-scale the $p_T$ values by a factor 1000 to end up between zero and one. The activation function is ReLU, except for the output layer. That one predicts the mean value $\langle p_T \rangle$ without any need for an activation function and the SoftPlus function for the error to have a smooth function which guarantees positive values for the error. We have checked that this setup with these hyper-parameters is not fine-tuned.

For the Bayesian network features we rely on Tensorflow Probability [57] with Flipout Dense layers [58] replacing the dense layer of the deterministic network. All networks are trained with the Adam optimizer [59] and a learning rate of $10^{-4}$, determined by early stopping when the loss function evaluated on the training dataset does not improve for a certain number of epochs. This patience was set to 10 for a training size of 1M jets and to larger values for smaller training sizes because the loss function is more fluctuating. For the Bayesian network with a training batch size of 100 we observe no over-fitting.

# 4 Momentum determination and statistics

As a first part of our Bayesian regression analysis we need to show how well the networks reconstructs the top momentum and what the limiting factors are. We then have to separate the statistical and systematic uncertainties. In analogy to Ref. [28] we first study how the size of the training sample affects the regression output, *i.e.* how well the Bayesian network keeps track of the statistical uncertainty.

To illustrate the output of our Bayesian network for a single jet we shoe an example in Fig. 2. Sampling from the weight distributions, $q(\omega)$, provides us with a Gaussian per sampled

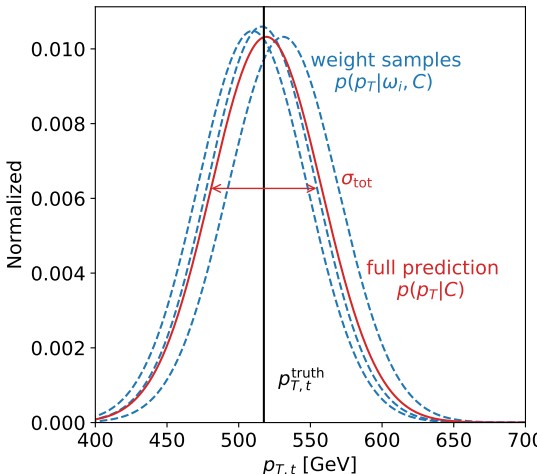

Figure 2: Illustration of the predicted distribution from our Bayesian setup for a single top jet. We show the individual predictions from sampling the weights (petrol) as well as the aggregate prediction (red) and the corresponding per-jet uncertainty $\sigma_{\text{tot}}$.

set of weights, shown in petrol. The combination of these distributions is shown in red. The width of the combined distribution is the predicted per-jet uncertainty $\sigma_{\text{tot}}$, defined in Eq.(5). For illustration purposes we pick a top jet where $p_{T,t}^{\text{truth}}$ coincides with the peak of the predicted distribution.

**Regression performance**

To begin, we show in the left panel of Fig. 3 the correlation between the measurable $p_{T,j}$ and the MC label $p_{T,t}^{\text{truth}}$. We see that over the entire range the two values are aligned well. This allows us to use $p_{T,j}$ as a proxy to the truth information, keeping in mind that we will eventually smear the truth label to describe the jet calibration. In the right panel of Fig. 3 we show the correlation between the central extracted $p_{T,t}$ value, which in Sec. 2 is properly denoted as the expectation value $\langle p_T \rangle$, and the label $p_{T,t}^{\text{truth}}$.

In the left panel of Fig. 4 we show the $p_{T,t}^{\text{truth}}$ distribution for the narrow slice $p_{T,j} = 600 \dots 620$ GeV. In the absence of initial state radiation the distribution is asymmetric. The simple reason is that the jet clustering can only miss top decay constituents, so we are more likely to observe $p_{T,j} < p_{T,t}^{\text{truth}}$. Aside from that we see a clear peak, suggesting that we can indeed represent $p_{T,t}^{\text{truth}}$ with $p_{T,j}$. Because the peak is washed out by ISR, we switch off ISR to make it easier to understand the physics behind our network task. In practice, this could be done through a pre-processing and grooming step.

Table 1: Performance of $p_{T,t}$ regression, uncertainty representing the standard deviation of 5 trainings. The narrow $p_{T,j}$ range refers to the 5k test jets, not the 500k training jets.

| $p_{T,j} = 600 \dots 620$ GeV | $\sqrt{\text{MSE}}$ | $\sqrt{\text{MSE}}/p_{T,j}$ | $\sqrt{\text{MSE}}$ | $\sqrt{\text{MSE}}/p_{T,j}$ |
|---|---|---|---|---|
| | With ISR | | Without ISR | |
| All jets | $69.7 \pm 0.2$ | $(11.43 \pm 0.03)\%$ | $50.6 \pm 0.1$ | $(8.30 \pm 0.02)\%$ |
| 75% most top-like | $67.8 \pm 0.2$ | $(11.11 \pm 0.01)\%$ | $45.5 \pm 0.1$ | $(7.47 \pm 0.02)\%$ |
| 50% most top-like | $66.5 \pm 0.1$ | $(10.89 \pm 0.01)\%$ | $41.8 \pm 0.1$ | $(6.85 \pm 0.01)\%$ |
| 25% most top-like | $66.5 \pm 0.1$ | $(10.89 \pm 0.02)\%$ | $40.4 \pm 0.1$ | $(6.63 \pm 0.02)\%$ |

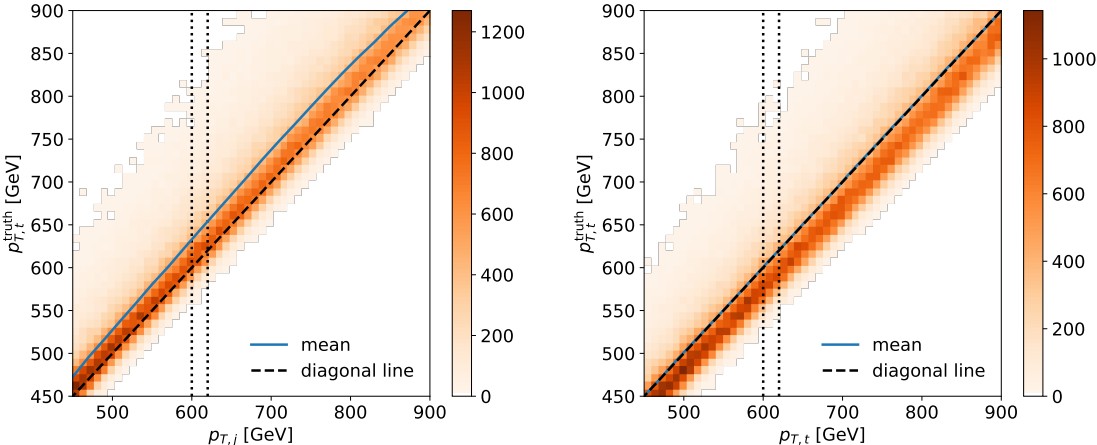

Figure 3: Correlation between the fat jet's $p_{T,j}$ and the truth label $p_{T,t}^{\text{truth}}$ (left) and between the extracted $p_{T,t}$ and the truth label $p_{T,t}^{\text{truth}}$ (right). Both correlations are shown with initial state radiation in the training and test samples switched off.

Whenever we have access to MC truth, we can measure the performance of the regression network for each top jet as $(p_{T,t} - p_{T,t}^{\text{truth}})^2$. The squared difference measure only uses the mean or central value reported by a Bayesian or deterministic network, not the additional uncertainty information from the Bayesian network. For a given test sample with $N$ top jets $t_i$ we construct the mean quadratic error as

$$\sqrt{\text{MSE}} = \left[ \frac{1}{N} \sum_{\text{jets } i} \left( p_{T,t_i} - p_{T,t_i}^{\text{truth}} \right)^2 \right]^{1/2}. \tag{14}$$

We evaluate it over homogeneous samples, for example our usual slice in $p_{T,j}$. In Tab. 1 we contrast results with and without ISR and show what happens if we limit ourselves to the most top-like jets based on a standard LoLa tagger [20], trained on events with ISR. To estimate the effect of different trainings we also give an error bar based on five independent trainings and the resulting standard deviation. Expectedly, the $p_T$-measurement benefits from more top-like events, but the effect is not as significant as in the HEPTOPTAGGER analysis [54]. One of the reasons is that we are using relatively large $R = 1.2$ jets for the high transverse momentum range. Similarly, we confirm that additional ISR jets have the potential to affect the top momentum measurement whenever hard extra jets enter the fat jet area.

In the right panel of Fig. 4 we show $\sqrt{\text{MSE}}$ as a function of $p_{T,j}$ for a bin width of 40 GeV. While the absolute error increases, the relative error on the extracted $p_{T,t}$ shrinks for more boosted jets. If we assume that an improved jet pre-selection can efficiently remove ISR contributions our regression network can measure the top momentum to roughly 4%. This result is only a rough benchmark to confirm that the regression network performs in a meaningful manner. It would surely be possible to improve the network performance, but we deliberately keep the network simple, to understand the way it processes information and the related uncertainties. From the right panel of Fig. 4 we know that boundary effects will appear already around 200 GeV away from the actual boundaries. Indeed, around $p_{T,j}$ we see such effects indicating the phase space boundary of $p_{T,j} < 1$ TeV in our training sample.

In the same Fig. 4 we also show this uncertainty estimate of the Bayesian network, $\sigma_{\text{tot}}$ as defined in Eq.(5). It follows the $\sqrt{\text{MSE}}$ estimate of the network error, indicating that the Bayesian output captures the same physics as the frequentist-defined spread of the central values.

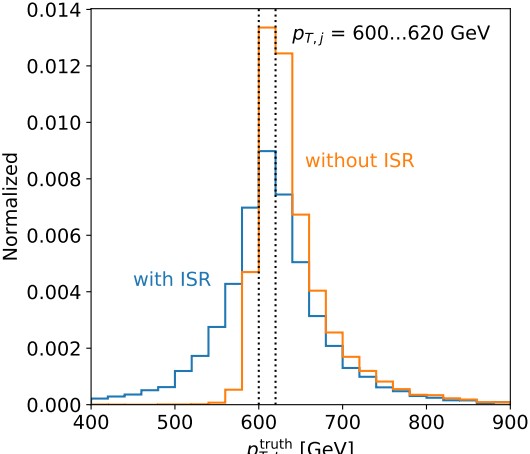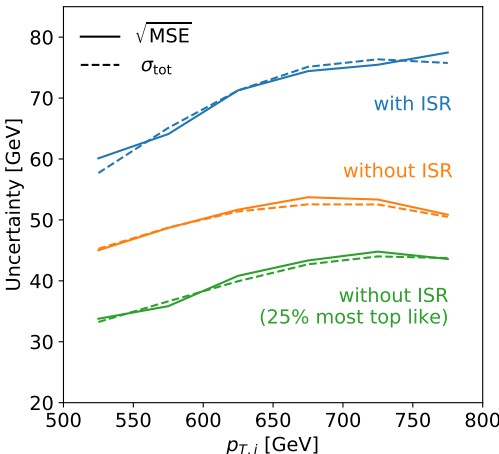

Figure 4: Left: distribution of the truth label $p_{T,t}^{\text{truth}}$ for jets with $p_{T,j} = 600 \ldots 620$ GeV, without and with initial state radiation. Right: regression uncertainty as a function of $p_{T,j}$ (solid), compared with the average $\sigma_{\text{stoch}}$ as the network output (dashed). The most top-like events are defined with a simple LoLa tagger [20].

**Training sample size and $\sigma_{\text{pred}}$**

As discussed in Sec. 2 the contribution $\sigma_{\text{pred}}$ to the uncertainty reported by the network can be identified as a statistical uncertainty in the sense that it should vanish in the limit of infinitely many training jets. In complete analogy to the classification task described in Ref. [28] we confirm this by training Bayesian networks on 2k, 5k, 10k, 15k, 20k, 30k, 50k, 100k, 200k, 500k, and 1M jets. We test these networks on the narrow range $p_{T,j} = 600 \ldots 620$ GeV, similar to the results shown in Tab. 1. The uncertainties quoted by the Bayesian network are shown in Fig. 5. In the lower part of the figure we first see that the statistical error $\sigma_{\text{pred}}$ indeed asymptotically approaches zero for 1M training jets. The error bars on the extracted uncertainty are given by the standard deviation of five independent trainings. As expected, they grow for smaller training samples, where the Bayesian networks also give fluctuating results.

In the same figure we also show the systematic $\sigma_{\text{stoch}}$ and the combined $\sigma_{\text{tot}}$, defined in Eq.(5). We confirm that the extracted $\sigma_{\text{stoch}}$ hardly depends on the size of the training sample. Once we have a reasonably number of training events it reaches a plateau of around 50 GeV or 8%, while for less than 10000 training events the network simply fails to capture the full information. We can compare the plateau value for $\sigma_{\text{stoch}}$ to the $\sqrt{\text{MSE}}$ value and find again that the two values agree. This allows us to conclude that $\sigma_{\text{stoch}}$ describes a systematic uncertainty and that it is related to the truth-based $\sqrt{\text{MSE}}$ estimate. We will discuss it in more detail in Sec. 5.

After observing the average effect of the training sample size on $\sigma_{\text{pred}}$ the obvious question is if we can understand this behavior. In the left panel of Fig. 6 we show the distribution of $\sigma_{\text{pred}}$ values for a sample of 400k jets. The network is trained on 100k jets with an extended range $p_{T,j} = 500 \ldots 900$ GeV. We see a clear maximum around $\sigma_{\text{pred}} \approx 5$ GeV, with a large tail towards large uncertainties. It is induced by the constraint that no network should quote an uncertainty close to zero.

The jet property we can relate to the $\sigma_{\text{pred}}$ behavior is the number of particle-flow constituents. As mentioned before, we cover up to 100 constituents for jets without ISR. Their effect on top tagging is discussed for instance in Ref. [20]. The center panel of Fig. 6 shows how the number of constituents in the test sample jets peaks at around 25, but with a tail

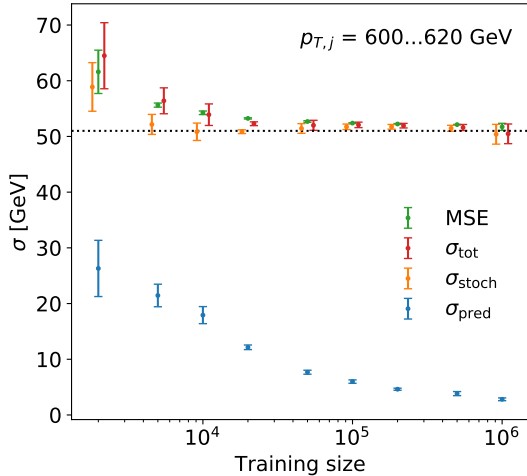

Figure 5: Uncertainty contributions $\sigma_{\text{pred}}$ and $\sigma_{\text{stoch}}$ as a function of the size of the training sample. The error bar represents the standard deviation of five different trainings. In addition we include $\sqrt{\text{MSE}}$ as defined in Eq.(14).

extending to 60. Jets with a larger quoted uncertainty have significantly more constituents. The same information is shown in the right panel, where we see the average number of jets increases with the range of quoted statistical uncertainties. The reason for this pattern is that also within the training sample the number of constituents will peak around 25, limiting the number of training jets with higher constituent numbers. We note that we could use the same argument using the jet mass.

**Frequentist approach**

From a practical point of view it is crucial to validate the Bayesian network using a frequentist approach. We do this by showing that predictions from many trainings of a deterministic network reproduce our Bayesian network results for the statistical uncertainty $\sigma_{\text{pred}}$.

For the deterministic networks we use the same architecture as for the Bayesian network. The loss function of the deterministic networks is the negative log-likelihood given in Eq.(9), and we fix the L2-regularization to match the Bayesian network in Eq.(8),

$$\lambda_{\text{L2}} = \frac{1}{2\sigma_{\text{prior}}N} \, , \tag{15}$$

where $N$ is the total training size and $\sigma_{\text{prior}} = 0.1$ is our prior width. We then train 40 deterministic networks on statistically independent samples, which we sample from the total of 2.2M training jets. Each set of deterministic network then predicts a mean and a standard deviation, in analogy to Eq.(11). The difference between the Bayesian evaluation and the frequentist networks is that we replace the integral over weights with a sum over independent networks.

For deterministic networks we need to avoid over-training. An over-trained set of networks will underestimate $\sigma_{\text{stoch}}$, while the spread represented by $\sigma_{\text{pred}}$ increases. However, it is not guaranteed that these two effects compensate each other for finite training time. This is why we introduce dropout for each inner layer with a rate of 0.1. This value is a compromise between network performance and over-training. Unlike in our earlier study [28] we do not use a MAP modification of the Bayesian network.

In Fig. 7 we compare the Bayesian and frequentist uncertainties for different training sample size. While the results agree well for properly trained networks or large training samples,

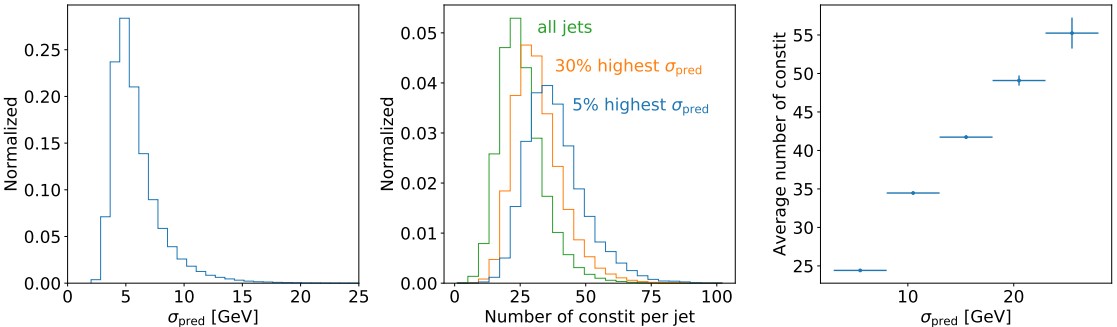

Figure 6: Left: distribution of the statistical uncertainty $\sigma_{\text{pred}}$ for 400k jets. Center: number of constituents per jet for different $\sigma_{\text{pred}}$. Right: average number of constituents per jet as a function of the extracted statistical uncertainty.

the frequentist approach slightly underestimates the uncertainty for small training samples. The plateau value of $\sigma_{\text{stoch}}$ depends on the chosen dropout value. Accounting for this effect we see that the training-size-dependent $\sigma_{\text{pred}}$ and the plateau value of $\sigma_{\text{stoch}}$, agree well between the Bayesian network and the frequentist sanity check.

## 5  Systematics and calibration

In our original paper [28] we have shown that the Bayesian setup propagates uncertainties from statistical and systematic limitations of the training data through a neural network. In addition to the usual output the Bayesian network provides event-by-event error bars. A limitation we encounter in Ref. [28] is that forcing the network output onto a closed interval, like a probability $p \in [0, 1]$, strongly correlates the the central value and the error bars in the network output. This makes it difficult to track systematic uncertainties.

We circumvent this problem by extracting the transverse momentum, which does not live on a closed interval. In the previous section this allowed us to decompose $\sigma_{\text{tot}}$ into a statistical component, $\sigma_{\text{pred}}$, and a systematic component, $\sigma_{\text{stoch}}$. What we still need to study is the actual output distribution of the Bayesian network, $p(p_T|M)$, and how it compared to the truth information from the test data.

**Variance of training data and $\sigma_{\text{stoch}}$**

In the upper left panel of Fig. 8 we show the correlation of $p_{T,t}^{\text{truth}}$ and $p_{T,j}$. The orange curves represent the maximum and the 68% CL interval in 20 GeV bin. The corresponding maximum and 68% CL interval of the BNN output are illustrated in blue. Both confidence intervals are constructed by requiring equal functional values at both ends. In the lower left panel we see why the two sets of curves agree very poorly: for the narrow $p_{T,j}$ slide the $p_{T,t}^{\text{truth}}$ distribution is all but Gaussian, while the Bayesian output in our naive approach is forced to be Gaussian, as seen in Eq.(11).

From Sec. 2 we know that it is not necessary to assume that the Bayesian network output is Gaussian. As a simple generalization we can replace the two-parameter Gaussian form of $p(M|\omega)$ in Eq.(10) with a mixture of Gaussians,

$$p(M|\omega) = \sum_i \alpha_{i,\omega} \, G(\langle p_T \rangle_\omega^{(i)}, \sigma_{\text{stoch},\omega}^{(i)}), \tag{16}$$

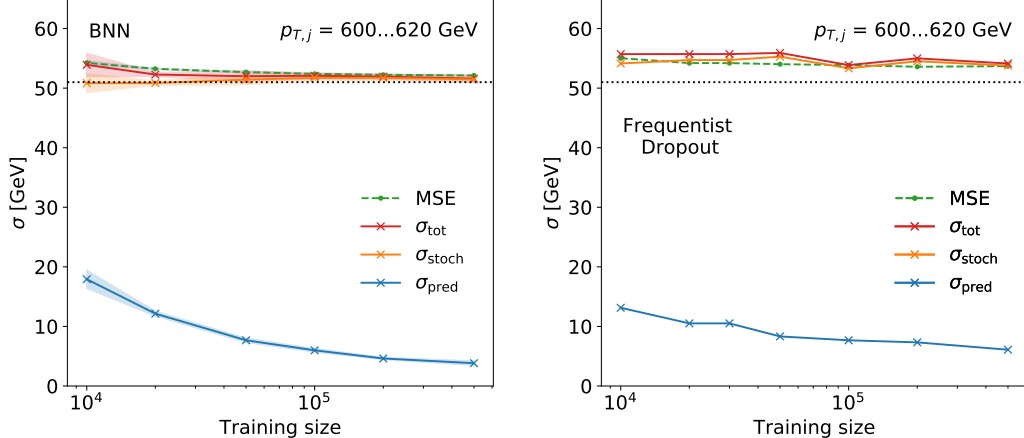

Figure 7: All uncertainties as a function of the training size, comparing the Bayesian network (left) with a (frequentist) set of deterministic networks (right). The left panel corresponds to Fig. 5, and the ranges indicate the standard deviation for five trainings.

with $\sum_i \alpha_{i,\omega} = 1$. The network output from Eq.(11) then becomes

$$
\text{NN}(\omega) = \begin{pmatrix} \alpha_{1,\omega} & \alpha_{2,\omega} & \cdots \\ \langle p_T \rangle_\omega^{(1)} & \langle p_T \rangle_\omega^{(1)} & \cdots \\ \sigma_{\text{stoch},\omega}^{(1)} & \sigma_{\text{stoch},\omega}^{(2)} & \cdots \end{pmatrix}. \tag{17}
$$

To guarantee $\sum_i \alpha_{i,\omega} = 1$ we use SoftMax as an activation function for $\alpha_{i,\omega}$ and the SoftPlus function for $\sigma_{\text{stoch},\omega}^{(i)}$ to ensure positive values. In the center and right sets of panels in Fig. 8 we see what happens if we use two or three Gaussians, specifically with the parameters averaged over weights and jets in a bin. For three Gaussians the BNN output and the $p_{T,t}^{\text{truth}}$ distribution agree perfectly. The corresponding parameters are shown in Tab. 2.

Technically, we follow Sec. 2 in extracting $\sigma_{\text{stoch}}$ and $\sigma_{\text{pred}}$ independently of the form of the underlying assumption. Two aspects render this computation slightly expensive: the integration over all weights and, if required, the combination of different predictions in one $p_{T,j}$ bin. On the other hand we know that $\sigma_{\text{pred}} \ll \sigma_{\text{stoch}}$ and we can always use narrow bin sizes. This means that in both cases we can replace the integrals by simply averaging over the parameters of the Gaussian mixture model. This implementation is computationally less expensive and gives us simple analytic expressions from which we extract the maximum and 68% CL interval.

**Noisy labels**

A crucial question in experimental physics is how we include a systematic uncertainty for instance on the jet energy scale in the training procedure. We can understand such an energy calibration when we remind ourselves that the jets in the calibration sample come with a measured reference value for their energies and the corresponding error bar; and that the calibration sample in our case is the training sample. There are two ways we can include the error on the calibration measurements in our analysis:

1A. fix the label or 'true energy' and smear the jets in the training sample;
1B. fix the jets and smear the continuous label in the training sample;
 2. train the Bayesian network on the smeared label-jet combination;
 3. extract a systematics error bar for each jet in the test sample.

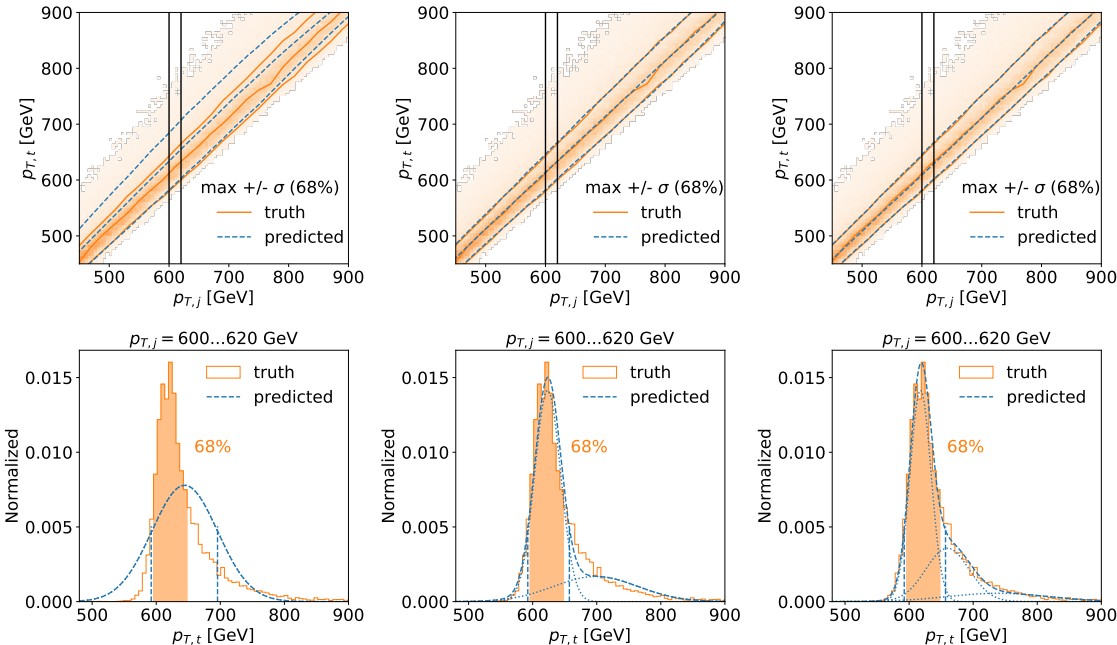

Figure 8: Upper: 2-dimensional distribution of $p_{T,t}^{\text{truth}}$ vs $p_{T,j}$ including its 68% CL around the maximum. In blue we show the BNN results. Lower: $p_{T,t}^{\text{truth}}$-distribution for a narrow slice in $p_{T,j}$. From left to right we approximate $p_{T,t}^{\text{truth}}$ with one, two, and three Gaussians.

In Ref. [28] we have followed the option 1A and encountered some practical/numerical problems when tracing the corresponding systematics to the network output. In this study we shift to the less standard and yet straightforward option 1B. We assume that jet calibration incorporates external information on the training sample, be it another measurement or a theory requirement (one-shell $Z$-decays) or a MC prediction. This information defines a label together with a corresponding error bar. This means we train our network on a fixed sample of jets with a smeared label representing the full reference measurement. In this approach we can trivially include additional uncertainties from pre-processing the training data, like running a jet algorithm of the $Z$-sample, removing underlying event and pile-up, etc. As a side effect our setup also allows us to capture possible transfer uncertainties, whenever our test sample cannot easily be linked to the training sample. In the ML literature such uncertainties are referred to as out-of-sample error.

To illustrate and test our setup we smear $p_{T,t}^{\text{truth}}$, the label in the training data, according to Gaussians with widths of

$$\sigma_{\text{smear}} = (4 \dots 10)\% \times p_{T,t}^{\text{truth}} . \tag{18}$$

In Fig. 9. we see that for a small amount of smearing the non-Gaussian shape of Fig. 8 remains, so we use two Gaussians in the BNN. For sizeable Gaussian smearing we see that the resulting distributions all assume a Gaussian shape and we can stick to the single-Gauss standard BNN. In both cases the distribution of the BNN output and the (smeared) label $p_{T,t}^{\text{truth}}$ agree almost perfectly.

From the previous sections we know that the reported uncertainty by the BNN includes a statistical uncertainty vanishing with an increasing amount of training data and a systematic uncertainty representing the stochastic nature of the training data. When we introduce another

Table 2: Parameters used in Fig. 8, specifically $p_{T,j} = 600...620$ GeV.

| | $\alpha^{(i)}$ | $\langle p_T \rangle^{(i)}$ | $\sigma_{\text{stoch}}^{(i)}$ | $\sigma_{\text{stoch}}$ | $\sqrt{\text{MSE}}$ | $\langle p_{T,t} \rangle$ | $\langle p_{T,t}^{\text{truth}} \rangle$ | Max | 68%CL | 68%CL (truth) |
|---|---|---|---|---|---|---|---|---|---|---|
| 1 | 1 | 644.4 | 51.43 | 51.4 | | 644.4 | | 644.4 | 593.0...695.9 | |
| 2 | 0.72 | 623.4 | 20.4 | 51.1 | | 644.1 | | 623.4 | 592.4...657.3 | |
| | 0.28 | 698.3 | 65.6 | | | | | | | |
| 3 | 0.59 | 617.8 | 16.6 | 51.5 | 52.2 | 643.8 | 643.8 | 619.1 | 592.4...656.8 | 590.0...654.0 |
| | 0.30 | 659.8 | 33.7 | | | | | | | |
| | 0.11 | 738.6 | 78.6 | | | | | | | |

uncertainty induced by smeared labels we expand Eq.(5) to

$$
\begin{aligned}
\sigma_{\text{tot}}^2 &= \sigma_{\text{stoch}}^2 + \sigma_{\text{pred}}^2 \\
&= \sigma_{\text{stoch},0}^2 + \sigma_{\text{cal}}^2 + \sigma_{\text{pred}}^2 \quad \Leftrightarrow \quad \sigma_{\text{cal}}^2 = \sigma_{\text{stoch}}^2 - \sigma_{\text{stoch},0}^2 ,
\end{aligned}
\tag{19}
$$

added in quadrature because of the central limit theorem. The baseline value $\sigma_{\text{stoch},0}$ is defined as $\sigma_{\text{stoch}}$ in the limit of no smearing. In Fig. 10 we show how $\sigma_{\text{cal}}$ correlates with the input $\sigma_{\text{smear}}$ over a wide range of scale uncertainties. As usually, the error bar represents the standard deviation from five independent trainings. This correlation shows that our network picks up the systematic uncertainties from smeared training labels perfectly. We note that, as before, this analysis does not require a Gaussian shape of the network output.

# 6 Outlook

We have shown that Bayesian networks keep track of statistical and systematic uncertainties in the training data and translate them into a jet-by-jet error budget for instance in a momentum measurement. Outside particle physics it is not unusual to treat uncertainties as a smearing of labels, whereas in particle physics we usually model them by smearing the input data. We show that smearing labels is a natural, feasible, and self-consistent strategy in combination with deep learning. An advantage of this approach is that the treatment of uncertainties is moved from the evaluation time to the training time and so-trained networks accurately report predictions of the central value as well as systematic uncertainties.

We have shown that the corresponding Bayesian networks allow us to cleanly separate statistical and systematic uncertainties. In addition, the smeared labels are ideally suited to translate uncertainties from reference or calibration data to the network output.

Technically, we have modified the Bayesian network approach of Ref. [28] to include non-Gaussian behavior. This step is crucial for modeling systematic uncertainties in general.

We emphasize that before this approach can be generally adapted, open questions such as multiple correlated uncertainties and the translation between input-uncertainties and label-uncertainties need to be answered. However, our first results show great promise for smeared labels describing uncertainties in particle physics applications of deep learning.

# Acknowledgments

We would like to thank Ben Nachman for many very useful discussions and Manuel Haußmann for getting us into Bayesian neural networks. ML is funded through the Graduiertenkolleg *Particle Physics Beyond the Standard Model* (GRK 1940). The authors acknowledge support

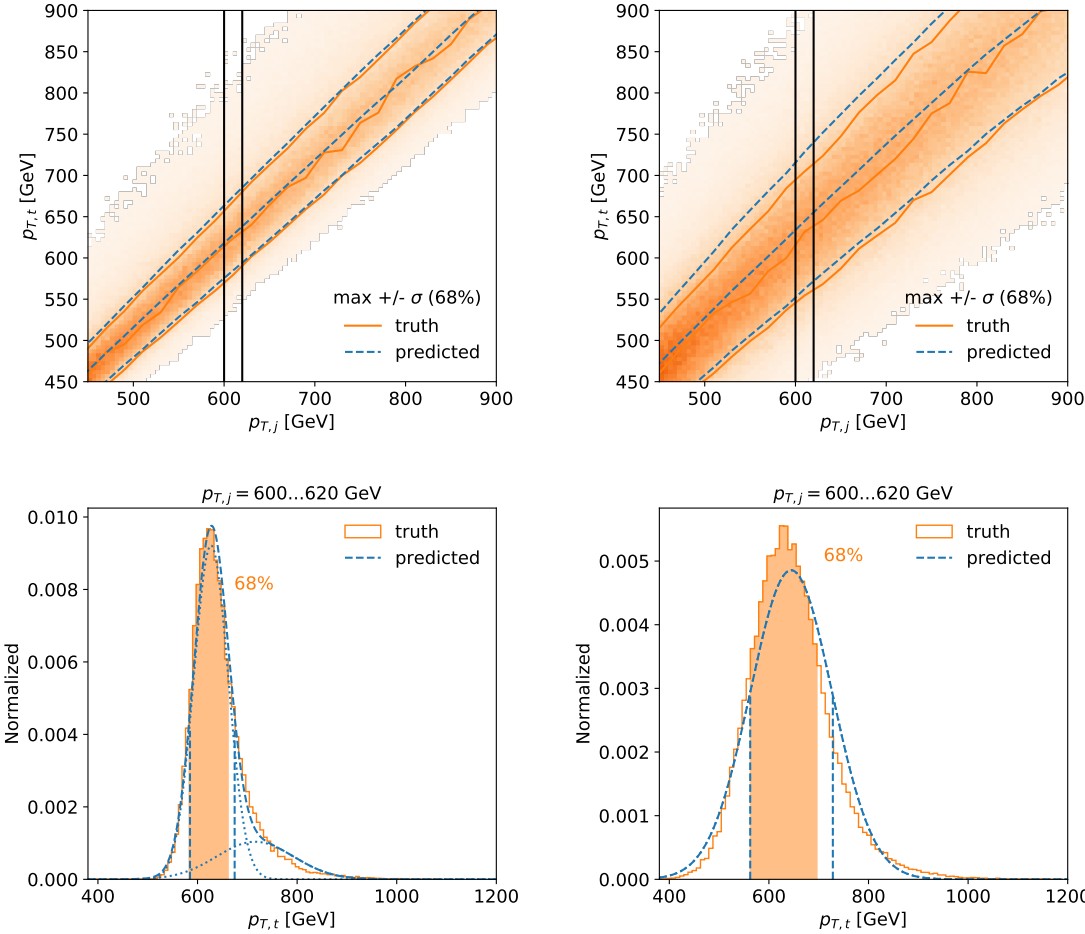

Figure 9: Upper: 2-dimensional distribution of $p_{T,t}^{\text{truth}}$ vs $p_{T,j}$ including its 68% CL around the maximum, after adding 4% (left) and 10% (right) Gaussian noise on the top momentum label. In blue we also show the BNN error estimate. Lower: corresponding $p_{T,t}^{\text{truth}}$-distribution for a narrow slice in $p_{T,j}$.

by the state of Baden-Württemberg through bwHPC and the German Research Foundation (DFG) through grant no INST 39/963-1 FUGG(bwForCluster NEMO). GK is acknowledges support by the Deutsche Forschungsgemeinschaft under Germany's Excellence Strategy - EXC 2121 *Quantum Universe* - 390833306.

# A    Comparison to smeared data

To further validate the proposed approach, Fig. 11 compares the performance of the BNN approach with a more traditional smearing of the input objects. For smearing the objects we use a Bayesian neural network trained on data without smearing and evaluate this network on a test dataset with modified inputs. Each jet in the test sample is smeared once up and once down, then the difference of the two network outputs is evaluated and divided by two. We then show the average in the given $p_{T,j}$-range. The BNN prediction is in good agreement with modified inputs, giving additional confidence in uncertainty predicted by the Bayesian network.

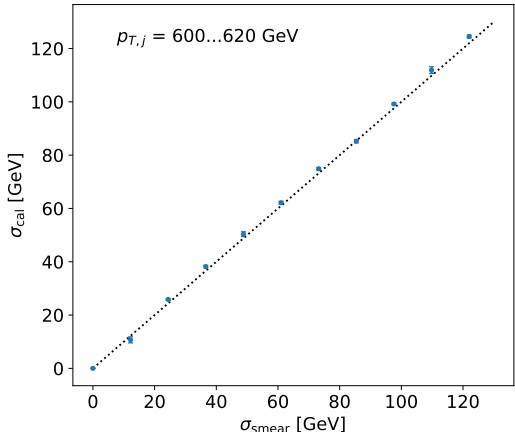

Figure 10: Correlation between $\sigma_{\text{stoch}}$, as given by the Bayesian network and the smearing $\sigma_{\text{smear}}$ applied to the label in the training data. The baseline $\sigma_{\text{stoch},0}$ is defined as $\sigma_{\text{stoch}}$ in the limit of no smearing. The error bars indicate the standard deviation from five independent trainings.

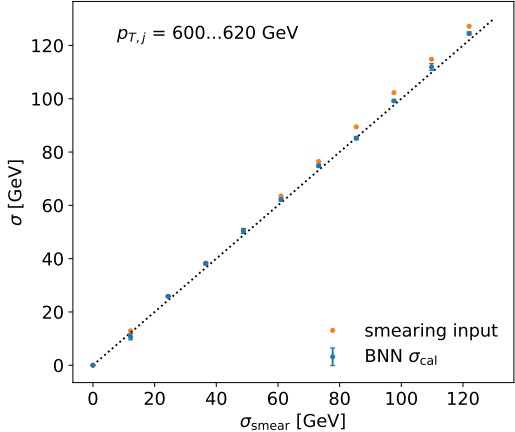

Figure 11: Comparison of the Bayesian approach, taken from Fig. 10 (blue), and smearing of input data (orange). When smearing the input data, we train a Bayesian network on nominal events and test it on inputs modified up and down by $\sigma_{\text{smear}}$.

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
