# Peer review of "Per-Object Systematics using Deep-Learned Calibration"

_SciPost Physics, doi:SciPost Phys. 9, 089 (2020)_

## Round 1 · Referee Report · Anonymous (Referee 1) · 2020-6-11

Strengths
Weaknesses
There are also countless typos throughout the paper, some of which significantly affect the meaning of the paper. For example on p 13 it is written "the p_{T,t}^truth distribution is all but Gaussian" but I think they meant it is "anything but Gaussian" or "not at all Gaussian".
Report
Requested changes
Specific critiques of section 2:
-
What is C?? As far as I can tell it is never explicitly defined. I thought it must be the truth label but at the beginning of section 2 it is called "an output" of the BNN which then confused me greatly.
-
What is the total loss function for the BNN? Is it just the L_{KL} or is there more?
-Why don't I see the truth label p^{truth}_{T,t} appear anywhere in the loss function?
-
Does the loss function for the BNN reduce to the usual mean-squared-error for regression tasks in some limit? If so how?
-
Related comment, the second term in eq (9) seems to have something resembling the usual MSE loss but it has the opposite sign.
-
Eq (3) made me think that <pT>omega was a derived quantity but then eq (10) tells me it's an output of the NN? Is the final layer of the NN performing the integration over pT? Similarly for sigma.
-
Is the loss (9) bounded from below? It comes from the KL divergence which is always nonnegative so I imagine it must be. However, it's not obvious from the form of eq (9). For example, why wouldn't the network choose <pT>omega very different from C and sigma->0 to make the second term of (9) arbitrarily negative?
-
More generally, can the authors provide more intuition as to the tradeoffs and tensions inherent in the loss function (9)? For example, what prevents the NN from simply choosing q(omega)=p(omega)?

---

## Round 1 · Referee Report · Anonymous (Referee 2) · 2020-6-15

Strengths
1- Reformulation of extraction of uncertainties as a regression task in a ML application
Weaknesses
1- Motivation is not entirely satisfying within the paper
2- Not written in a way to be self-contained
3- Ultimately an application of a statistical technique to physics, but unclear what physics one learns from it
Report
This would also improve the readability of the paper. The authors state that this paper is a continuation of their reference [28], but those results aren't really reviewed or placed in context in the current paper. Just from this paper alone, it's not entirely clear what the weaknesses or shortcomings of their earlier work was. Perhaps that's not relevant for this paper, but it could be placed in context better for the reader who is less familiar with all of the literature.
Requested changes
1- At the end of the first paragraph, the authors state that "By now we can consider top jet classification at the level of tagging performance as essentially solved [26,27], giving us room to consider other open question in machine learning and jet physics." I do not know what the authors mean by "essentially solved". While ML techniques have demonstrated good discrimination power in the problem of top tagging, this does not mean that it is a "solved problem". In particular, it is still unknown what physics or properties of jets is responsible for this discrimination power. Again, just demonstrating discrimination with a statistical technique is not physics. The authors should re-phrase this sentence, eliminating the implication that top tagging is "solved".
2- I was a bit unclear about what the object C is in section 2. C is the output of the machine, which is some vector function of the inputs, right? If this is true, then I do not understand equation 8 in which a number (<pT>) is subtracted from C. If C is not a vector, then the authors need to state what it is clearly.
3- Further, in Eq. 8, I think the signs on the right side of the equation are incorrect. As a normalizable Gaussian, everything on the right of the equation should be negated.
4- At the end of the first paragraph on page 8, the authors state that they could control ISR by "pre-processing and pruning step." I think the authors mean more general "grooming" and not "pruning" here, because "pruning" is a specific contamination-removal technique. If the authors do mean pruning, then they should add a reference to the original literature on pruning.
5- The caption of Table 1 is awkward. The first-person singular is used; however, the author list is multiple people. This caption should either be re-written in passive voice or use first person plural (i.e., we).

---

## Round 2 · Referee Report · Anonymous (Referee 2) · 2020-11-7

Report

The authors have satisfactorily addressed the points in my previous review.

---

## Round 2 · List of Changes

• section 2 was rewritten
  • minor changes in other sections

---

## Editorial Decision

published